# Households' Willingness to Pay for Renewable Energy Alternatives in Thailand

Surasak Jotaworn [1], Vilas Nitivattananon [2,*], Ornuma Teparakul [3], Thanakom Wongboontham [2], Masahiro Sugiyama [4], Masako Numata [4] and Daniel del Barrio Alvarez [5]

[1] Department of Social Science, Faculty of Liberal Arts, Rajamangala University of Technology Thanyaburi, 39 Moo 1, Klong 6, Khlong Luang, Pathum Thani 12110, Thailand; surasak_jo@rmutt.ac.th

[2] Urban Innovation and Sustainability, Department of Development and Sustainability, School of Environment, Resources and Development, Asian Institute of Technology, Pathum Thani 12120, Thailand; twongboontham@gmail.com

[3] Faculty of Sociology and Anthropology, Thammasat University, Rangsit Center, Khlong Nueng Subdistrict, Khlong Luang District, Pathum Thani 12120, Thailand; t.ornuma@gmail.com

[4] Institute for Future Initiatives, The University of Tokyo, Tokyo 113-0033, Japan; masahiro@ifi.u-tokyo.ac.jp (M.S.); m.matsuo-numata@05.alumni.u-tokyo.ac.jp (M.N.)

[5] Department of Civil Engineering, The University of Tokyo, Tokyo 113-8656, Japan; danieldelbarrioalvarez@g.ecc.u-tokyo.ac.jp

* Correspondence: vilasn@ait.asia

**Abstract:** While the problems about the environmental effects of traditional energy use are growing, Thailand has a rapid response by increasing its renewable energy (RE) policy. Even though Thailand has seen rapid growth in RE, it has been focusing on supporting the producers and not considering the users. In addition, there were few studies on RE receivers in Thailand. To reach sustainable growth and increase the empirical study, this research aims to analyze the socio-economy, electric consumption behavior, attitude, opinions, and cognition of households in Bangkok Metropolitan to willingly pay for RE alternatives in Thailand. A questionnaire survey was carried out for 250 households covering six administrative districts, selected through multistage and stratified sampling techniques. The data were analyzed by descriptive statistics and conditional logit regression. It is found that the overall household in Bangkok still unchanged the status of electricity production based on the findings of socio-economy, behavior, and psychological factors. Considering to pay for RE alternatives, households are willing to pay (WTP) for solar energy at the highest level among other types, and biomass is the least willing to pay when the RE share is expected to reach 40%. These results are relevant for the planning of RE in the metropolitan region and the methodology applicable to other regions for extending RE opportunities to the national level.

**Keywords:** choices; households; metropolitan; renewable energy; willingness to pay

## 1. Introduction

It has been widely understood that efforts to tackle climate change are urgently necessary, as evidenced by the universal enactment of the Paris Agreement. The energy usage of all industries should reduce greenhouse gas emissions. The publication "Net Zero by 2050: A Roadmap for the Global Energy System" by the International Energy Agency (IEA) outlines a comprehensive strategy for achieving worldwide net-zero emissions by 2050, as documented in the IEAs 2021 report (IEA 2021a). As stated in this report, there is a significant difference between where we should be in terms of greenhouse gas emissions to meet the target and where we are currently (IPCC 2022). As a result, public acceptance of energy pricing will be critical for progress in strategically lowering $CO_2$ and greenhouse gas emissions from fossil fuels.

While problems about the environmental effects of traditional energy use are growing, the prospect of creating clean and sustainable energy from renewable energy (RE) sources

is generating excitement across the world (ERIA 2019). RE sources emit fewer harmful pollutants and lower emission rates than fossil fuels. Even though several countries have emphasized the need for coal as a diverse source and to avoid over-reliance on natural gas. Higher levels of renewable deployment should be considered since they are indigenous sources of supply, and if the goal is $CO_2$ emission reduction, then renewable energy is the better choice (IRENA 2017).

In the twenty-first century, electric power has become the most important source of energy for daily living. According to the growth of global electricity consumption, per capita electricity consumption in developing countries will double by 2030, reaching nearly 2400 kWh per person, while developed countries will increase by 7% (European Environment Agency 2015). Electricity is a secondary resource generated from a mix of natural sources for electricity production. Unlike renewable resources, they are finite and will deplete over time. Therefore, a continuous and consistent supply must be maintained to meet our energy needs. The degree of willingness to pay (WTP) for renewable energy is an important indicator of how to respond to increased renewable installations (REI 2020).

Thailand's power system is characterized by a high proportion of natural gas-fired production capacity (about 60% of installed capacity), hydropower generation with storage, and a minor proportion of variable renewable energy, less than 4 percent. (IRENA 2017). Based on that small percentage, the total solar power output is 19–20 MJ/m$^2$ per day. Thailand is now ranked fourth out of six countries, trailing only the United States of America (Ministry of Energy 2015). The Alternative Energy Development Plan (AEDP) policy for the period 2015 to 2036 seeks to construct an extra 7.5 gigawatts (GW) of variable RE capacity by 2036, primarily from solar photovoltaics, to meet Thailand's peak energy demand of roughly 30 GW in 2015 (DEDE 2021). It is necessary to create an incentive scheme in Thailand that allows companies to generate renewable energy and purchase it back from the government's electric authority at a discounted rate known as Adder (feed-in premium), making Thai entrepreneurs interested in renewable energy, particularly solar energy. As a result, several new vendors have sprung up in Thailand (Suanmali et al. 2018).

Even though Thailand has rapid growth in renewable energy because of the mentioned supporting policy, it was just supporting the power generators or supply side, not realizing the capability of power receivers. Thus, the key demand motivation of households' level to adapt renewable energy for consumption is to determine whether they are willing to pay for it (Pattanayak et al. 2006). The research question has arisen: what is the electricity consumption behavior of people living in metropolitan areas? What are their attitudes and opinions toward renewable energy? Are they willing to pay for renewable energy? To reach a valid answer to these questions, this research aims to analyze the electric consumption behavior, attitude, and opinions toward RE and the cognition of environmental problems of households in metropolitan areas to apply RE in the future, in the case of Bangkok Metropolitan. The findings are expected to contribute to the planning of renewable energy resources in metropolitan areas by approving the main hypothesis that households in Bangkok with a strong presence in social, economic, and environmental dimensions demonstrate a willingness to pay for renewable energy. In addition, the approach that may be used in other locations to expand renewable energy potential to the national level.

## 2. Literature

### 2.1. Willingness to Pay

Willingness to pay (WTP) is the monetary sum an individual is prepared to spend on a particular product or service. It serves as a means of gauging the financial worth attributed by a specific demographic to a given product or service. WTP values can furnish insights for quantifying both tangible and intangible aspects, which may not currently have a presence in the market. This approach was first used in the field of environmental economics to assess the monetary worth of environmental concerns, and healthcare services, and to measure public preferences, and it also helps in decision-making processes (Piran et al. 2001). It has found application across numerous domains, encompassing project feasibility,

tariff establishment, policy formulation, and cost-benefit analysis (Gunatilake et al. 2007). The assessment of Willingness to Pay (WTP) yields an enhanced pricing structure, offering the utmost potential profit margin for any product or service. Consequently, it paves the way for optimizing both sales volumes and profit margins (Chantana et al. 2021). In addition, there were very few studies on WTP for RE (IEA 2021b), and WTP results through the Discrete Choices Experiment (DCE) analysis in Thailand. To reach a sustainable growth of renewable energy, an analysis of people's WTP is needed to study.

## 2.2. Discrete Choices Experiment (DCE)

Recent progress in the realm of Discrete Choice Experiment (DCE) theory and methodologies has been greatly influenced by the contributions of McFadden. He expanded Thurstone's initial theory of paired comparisons, which involved evaluating pairs of choice alternatives, to encompass scenarios with multiple comparisons (McFadden and Train 2000). Specifically, Random Utility Theory (RUT) posits the existence of an unobservable latent construct referred to as "utility" residing within an individual's cognitive domain, which remains beyond the purview of empirical observation by researchers. In other words, individuals possess a "utility" value for each available choice alternative, yet these utilities remain inherently unobservable to researchers. Within the framework of RUT, systematic components encompass attributes that elucidate distinctions among choice alternatives, while covariates elucidate variations in individuals' choices (Louviere et al. 2010).

The utilization DCE presents a more accessible avenue for the estimation of the intrinsic worth of specific attributes comprising an environmental entity, such as a landscape. This holds significance due to the prevalent focus within management decisions on modifying attribute levels rather than the wholesale gain or loss of the entire environmental asset. DCE offers the advantageous capability of discerning the marginal values associated with attributes, a task that may prove challenging when relying solely on revealed preference data, primarily owing to issues like co-linearity or a dearth of variability. Consequently, DCE may exhibit certain merits over the Contingent Valuation Method (CVM) with regard to benefit transfer, provided that environmental assets can indeed be deconstructed into quantifiable attributes possessing monetary values amenable to estimation. This holds true, especially when the DCE models encompass pertinent socioeconomic variables. The recurrent sampling methodology employed in DCE enables the conduct of internal consistency assessments, wherein models can be constructed and evaluated on subsets of the available data (Hanley et al. 1998).

## 2.3. Energy Situation in Thailand

As previously stated, a significant component of Thailand's power system is based on natural gas-fired electrical generation, with a tiny fraction based on renewable energy. Thailand's latest Power Development Plan (2018) intends to expand the amount of generating capacity powered by renewable energy sources to 36% by 2037. Due to technological advancements and quick cost reductions, the country is seeing a strong uptake of variable renewable energy (VRE), notably solar PV (Tongsopit and Greacen 2013). Thailand is also regarded as a REIa leader in the field of renewable energy development. Thailand was one of the first Asian countries to use a feed-in tariff (FIT) system. When premium rates are placed on top of wholesale power prices, the FIT, also known as the Adder Program, entered into force in 2007. In 2013, the plan was changed to a fixed FIT (UNESCAP 2020). The continuous rise of renewable energy in its power mix has been observed in recent years as a result of well-balanced and responsive regulations (Malahayati 2020).

Noted: Imports encompass foreign hydropower and lignite, while renewable energy sources include wind, solar PV, and bioenergy. These data sources are derived from the Energy Policy and Planning Office (2020) and EGATs Electricity reports.

As depicted in Figure 1, natural gas has maintained its position as the predominant source of power generation in Thailand over the past two decades. It constituted approximately 70% of the total power generation in the early 2000s. The generating mix has

gotten increasingly diverse in recent years, with the percentage of renewables and imports growing in 2019. Renewable energy's percentage of overall generation has consistently climbed in recent years, going from 12% in 2017 to nearly 20 percent in 2019. Hydropower (both local and imported) accounts for the majority of renewable energy, with solar and wind power accounting for around 4% of total output.

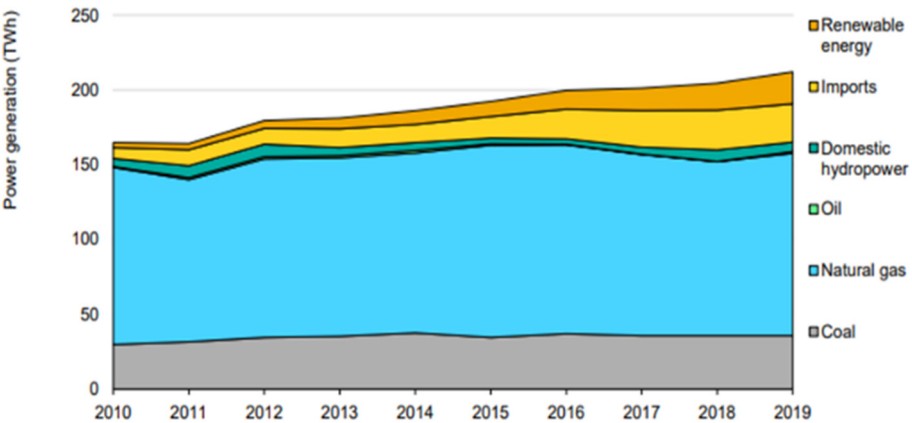

**Figure 1.** Thailand's power generation by fuel type, 2010–2019 (IEA 2021b).

Despite Thailand's steady progress and its emerging leadership in the field of renewable energy (RE), the country continues to exhibit a significant dependence on oil and natural gas. The Thai government has taken proactive measures to rectify this situation, including the introduction of the Alternative Energy Development Plan (AEDP), aimed at boosting the utilization of renewable energy sources. Additionally, the 20-Year Power Development Plan 2010–2030 outlines an ambitious target to reduce gas consumption by approximately 12.6% by 2030 while simultaneously promoting the adoption of more renewable energy sources and nuclear power (Malahayati 2020). Thailand, as a middle-income country, mostly allocates budgets for economic development, including the supply of water and electricity for several economic activities such as agriculture, industry, tourism, and SMEs (Nitivattananon and Sa-nguanduan 2013). To respond to the intention of the Thai government, it is necessary to realize the acceptance cost and price of adapting the RE at the household level. The benefit of recognizing this data are that it will assist policymakers and the government in planning and issuing strategies to increase the trend of RE use; it becomes potential bottom-line data. Moreover, there is a limited study on WTP for renewable energy in developing countries and Thailand. This research broadens the academic application of the WTP idea to underdeveloped areas.

*2.4. Willingness to Pay for Renewable Energy in Thailand*

WTP measurement is a valuable tool in economics and market research that helps assess the value individuals or consumers place on a particular product, service, or attribute. It helps in devising effective pricing strategies by understanding what customers are willing to pay. Companies can set prices that are competitive yet profitable, and they can tailor pricing to different market segments. Meanwhile, the price of electricity in Thailand varies depending on the Government. Thailand's trajectory is progressively inclining, positioning itself as a prominent figure in the realm of renewable energy (RE). The Thai government has unveiled the Alternative Energy Development Plan (AEDP) as a strategic initiative aimed at augmenting the utilization of renewable energy sources. Furthermore, the 20-Year Power Development Plan for the period 2010–2030 has been established with the overarching goal of diminishing the nation's reliance on natural gas by approximately 12.6% by the year 2030 while concurrently fostering the integration of a diversified portfolio of renewable energy resources. However, it is important to note that accurately measuring WTP can be challenging, and the results may vary depending on the methodology and

context of the RE situation in Thailand. Moreover, there was one article that used the same tools and method to find WTP for renewable energy in Myanmar. The result found that people in Myanmar were willing to pay for renewable energy for each type of renewable electric power generation source. Global advancements in the adoption of solar power have paralleled the recent decline in prices, as noted by Numata et al. in 2021. Based on the problem statement and a review of the literature, it carefully formulated the following hypothesis: Households in Bangkok with a strong presence in social, economic, and environmental dimensions demonstrate a willingness to pay for renewable energy.

## 3. Method

The quantitative approach was used to analyze WTP for renewable energy in households in the metropolitan area of Bangkok, Thailand. This study specifically investigates the WTP of renewable energy in Thailand to respond to the intention of the Thai government through the discrete choice experiment (DCE).

### 3.1. Overall Step and the Main Technique

Such methodologies were initially used in the field of environmental economics to evaluate the valuation of environmental concerns and preferences, and they also quantitatively enhance decision-making. In order to respond to these estimates, the overall structure of the questionnaire was developed to cover socio-economic aspects (i.e., gender, occupation, education, age, and income), electricity consumption, attitudes and opinions about renewable energy, and DCE. There were five steps to the overall methodology:

- Step 1: Develop the concept of WTP and DCE to cover all aspects of this study.
- Step 2: Design the random sampling strategy for the Bangkok Metropolitan area.
- Step 3: Survey the planned households in Bangkok.
- Step 4: Analyze the WTP of households in Bangkok.
- Step 5: Discuss the findings and recommendations from those results.

This study is focusing on step 4, which is a specific analysis of WTP through a DCE survey of households in metropolitan areas. DCE is the most widely used method for evaluating WTP for an environmental good or service in different alternatives to public goods (Xie and Zhao 2018). Hence, this study used a similar approach to DCE from Numata et al. (2021). However, to support the Thai government's intention, the socio-economic component of the inquiry was the first element of the research's structure, which was designed to provide a comprehensive understanding of the target sample's characteristics. The electricity consumption behavior was the second examination, followed by the attitude toward renewable energy and the cognition of environmental issues.

### 3.2. Study Area

Bangkok encompasses a total area of 1569 square kilometers, accommodating a population of 5.6 million individuals and comprising 2.8 million registered households (HHs). The city is subdivided into 50 districts and is geographically separated by the Chao Phraya River, resulting in the regions of Bangkok and Thonburi. The governance of Bangkok falls under the jurisdiction of the Bangkok Metropolitan Administration (BMA), which operates in accordance with the Bangkok Metropolitan Administration Act of 1985, holding responsibility for the overall management of the city (BMA 2016).

Figure 2 displays that the BMA divides the city into 6 administrative zones: Central Bangkok (a yellow color that comprises 9 districts), South Bangkok (an orange color that comprises 10 districts), North Bangkok (a light blue color that comprises 7 districts), East Bangkok (a light brown color that comprises 9 districts), North Thonburi, and South Thonburi (a pink color that comprises 8 and 7 districts).

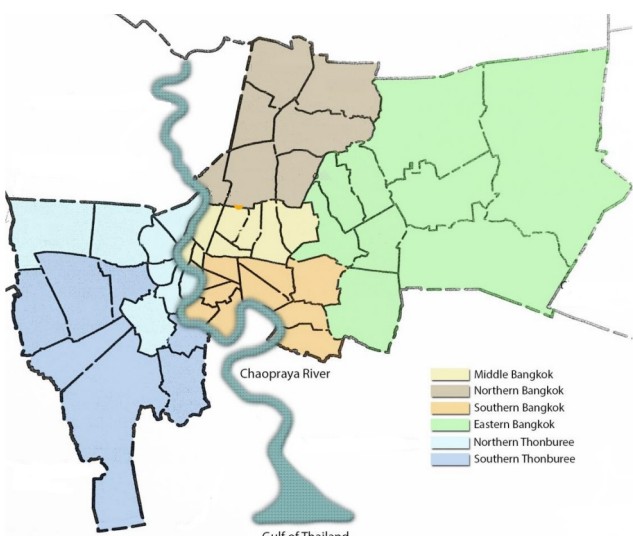

**Figure 2.** Administrative zones of the Bangkok Metropolitan Administration. Source: Rujibhong et al. (2018).

### 3.2.1. Selection Areas for Survey

Multi-Stage and stratified sampling techniques were applied to this study to ensure the random selection of districts, sub-districts, and households, respectively (Terris-Prestholt et al. 2019). Details are shown in Table 1.

**Table 1.** List of Sampled Districts and Sub-Districts within Each Zone.

| Zone | District | Sub-District | Main Road | Character |
|---|---|---|---|---|
| Central Bangkok | Ratcha The | Thanon Phyathai Thanon Petchaburi | Phyathai Petchaburi | Commercial Building District Commercial Building District |
| | Din Daeng | Din Daeng Ratchadaphisek | Asoke-Din Daeng Ratchadaphisek | Residential Commercial Building District |
| South Bangkok | Wattana | Klong Toei Nue Klong Tan Nue | Asoke Mantri Sukhumvit | Commercial Building District Commercial Building District |
| | Bang Na | Bang Na Nue Bang Na Tai | Sukhumvit Bang Na Trad | Residential Residential |
| North Bangkok | Laksi | Tung song hong Talad Bangkhen | Ngam Wong Wan Chaeng Wattana | Residential Residential |
| | Bang Khen | Anusawari Tha Reang | Ram Indra Ram Indra | Residential Residential |
| East Bangkok | Lad Krabang | Klong Songtonnoon Thab Yao | Sri-Nakharin Romklao Pracha Pattana | Industrial and residential Industrial and residential |
| | Prawet | Prawet Nong Bon | Pattanakarn Sri-Nakharin | Suburban and residential Suburban and residential |
| North Thonburi | ThaweWattana | Thawee Wattana Salathamm | Putthamonton sai 3 Putthamonton sai 2 | Suburban and agriculture Suburban new residential |
| | Taling Chan | Taling Chan Chim Plee | Ratchapruk Putthamonton sai 1 | Suburban and agriculture Suburban and agriculture |
| South Thonburi | Bang Khae | Bang Phai Lak Song | Putthamonton sai 2 Petchkasem 69 | Suburban and agriculture Suburban and agriculture |
| | Nong Khem | Nong Khem Nong Khang Plu | Liab Klong Phasi Charoen Putthamonton sai 3 | Suburban and new residential Suburban and new residential |

Table 1 is based on the sampling technique as follows: Stage (1) Stratified Sampling. Bangkok is divided into 6 administrative zones: Central Bangkok, South Bangkok, North Bangkok, East Bangkok, North Thonburi, and South Thonburi. Stage (2) Simple Random

Sampling to select two districts from each zone. Stage (3) Simple Random Sampling to select two sub-districts from each district. Stage (4) Select the main road in each sub-district and perform Area Based Sampling to select households, starting from the first alley of the road. Selected households must locate at least 3–5 houses apart.

### 3.2.2. Identification of Total Samples and Allocation among Different Zones/Sub-Areas

The sample size needed to calculate WTP using a DCE varies depending on the investigation. According to Numata et al. (2021), a sample size of 250 can be statistically analyzed using Equation (Numata et al. 2021).

$$\frac{nta}{c} > 500$$

where '$n$' represents the total number of respondents, '$t$' stands for the total number of tasks, '$a$' denotes the total number of alternatives, and '$c$' signifies the total number of attribute levels, our design parameters are as follows: '$c$' is capped at 5 (at maximum), '$t$' is set at a maximum of 8, and '$a$' equals 2 because the "status quo" alternative is excluded from the count. Based on these considerations, we determined that the number of respondents, denoted as '$n$', should exceed 156. Consequently, we collected a substantial sample of 250 responses for each type of information material. The sampling allocation of the total samples of 250 targeted households was proportionally allocated to zones, districts, and sub-districts.

Table 2 shows the sampling allocation and allocating proportion to zones, districts, and sub-districts. Sampled households will be randomly selected on the allocated target main road in each of the selected sub-districts.

**Table 2.** Number of Required Samples in each Sub-District.

| Zone | District | Sub-District | HHs Sub-District | % Weight | n for Sub-District |
|---|---|---|---|---|---|
| Central Bangkok | Ratcha Thewi | Thanon Phyathai | 11,621 | 13.76 | 4 |
| | | Thanon Petchaburi | 11,607 | 13.75 | 4 |
| | Din Daeng | Din Daeng | 37,059 | 43.89 | 14 |
| | | Ratchadaphisek | 24,151 | 28.60 | 9 |
| South Bangkok | Wattana | Klong Toei Nue | 17,301 | 13.69 | 6 |
| | | Klong Tan Nue | 38,668 | 30.60 | 14 |
| | Bang Na | Bang Na Nue | 33,350 | 26.39 | 12 |
| | | Bang Na Tai | 37,060 | 29.32 | 13 |
| North Bangkok | Laksi | Tung song hong | 39,641 | 23.32 | 11 |
| | | Talad Bangkhen | 17,742 | 10.44 | 5 |
| | Bang khen | Anusawari | 59,025 | 34.72 | 17 |
| | | Tha Reang | 53,589 | 31.52 | 15 |
| East Bangkok | Lad Krabang | Klong Songtonnoon | 32,784 | 29.71 | 16 |
| | | Thab Yao | 17,825 | 16.15 | 9 |
| | Prawet | Prawet | 35,922 | 32.55 | 17 |
| | | Nong Bon | 23,813 | 21.58 | 12 |
| North Thonburi | ThaweWattana | Thawee Wattana | 8973 | 16.39 | 5 |
| | | Salathamm | 24,922 | 45.51 | 14 |
| | Taling Chan | Taling Chan | 11,351 | 20.73 | 6 |
| | | Chim Plee | 9516 | 17.38 | 5 |
| South Thonburi | Bang Khae | Bang Phai | 14,597 | 14.43 | 6 |
| | | Lak Song | 24,661 | 24.38 | 11 |
| | Nong Khem | Nong Khem | 30,333 | 29.99 | 12 |
| | | Nong Khang Plu | 31,549 | 31.19 | 13 |

### 3.3. Collection Process, Tool Development, and Analysis

### 3.3.1. Survey Process

The research team played pivotal roles in gathering household data, following a structured 5-step data collection process:

1.  Initial Household Selection: The researchers began by heading to the designated main road and selecting the first household in a random manner.
2.  Introduction and Invitation: After identifying a household, the researchers introduced themselves to the resident and extended an invitation to participate in the survey.
3.  Consent Information: Subsequently, the researchers presented the consent information contained within the questionnaire to the respondent, ensuring ethical approval for their participation.
4.  Collection of Electricity Bill Data: The researchers proceeded to request access to and capture a photograph of the most recent electricity bill as a part of the survey.
5.  Conclusion and Next Household Selection: Upon completing the survey, the researchers provided any agreed-upon incentives to the respondent. They then requested permission to take a photo and, following that, proceeded to select the next household randomly. It is important to note that if a respondent declined to participate at any point during the aforementioned steps, the enumerator would move on to the next household selected randomly in accordance with the survey protocol.

### 3.3.2. Questionnaire Development

Based on the theoretical background, the DCE survey was used to identify people's stated preferences and examine the major indications of willingness to pay. The DCE method, which is also based on questionnaires, evaluates the value of the environment by asking for preferences for various alternatives to using power from several sources of renewable energy. Respondents were asked which of several alternatives they preferred. DCE is the most widely used technique for evaluating WTP for an environmental good or service. In this study, DCEs present respondents with a series of choice alternatives where they are asked to choose between two or more alternative goods or services with simultaneously varying attribute levels (Numata et al. 2021).

Table 3 shows that the value and sources of renewable energy in the three alternatives to the choice task (CT) were different. When making these choices, respondents find themselves in a situation where they must weigh the advantages and disadvantages of the different attribute levels presented within each alternative service. Through the application of econometric analysis to the choices made by respondents, it becomes possible to extract the value or utility associated with each attribute level in relation to the others, as highlighted in the work of Danne et al. (2021).

**Table 3.** Sample discrete choice experiment (DCE) survey question.

| Choice Task1 | Alternative A | Alternative B | Alternative C (Status Quo) |
| --- | --- | --- | --- |
| % Renewable Energy Share | 35% Renewable Energy  | 15% Renewable Energy  | 9% Renewable Energy  |
| Main Type of renewable Energy |  Biomass |  Hydro |  Solar |
| % Increase in Monthly Electricity Bill | Your monthly electricity bill will increase by 25% | Your monthly electricity bill will increase by 2% | No change |

Source: (Numata et al. 2021).

### 3.3.3. Attributes and Levels

The questionnaire survey was developed by many experts and scholars. The design of the questionnaire was adapted from previous studies in Thailand and International research. The varied values in the three alternatives of the DCE survey were determined after a conversation with those specialists based on past analysis and Thailand's conditions: share of renewable energy within all-electric power sources in 2030, type of renewable energy, and rate of increase in electricity charges (Numata et al. 2021).

Table 4 shows the relevant higher value based on the current rate of renewable energy in Thailand (Figure 1) and the expected plan. In the year 2019, Thailand boasted a total installed generation capacity of 47 GW. This capacity was distributed among various sources, with 30 GW originating from gas-fired power plants, 6 GW from coal-fired facilities, and 11 GW attributed to renewables. It is worth noting that the renewable category encompasses hydropower capacity situated in neighboring countries but directed towards serving Thailand's energy needs. As per the Thai government's projections, they anticipate a notable shift in the energy landscape by 2030. Specifically, they aim to increase the proportion of renewables in total electricity generation to approximately 25%. An alternative scenario proposes a more ambitious target, envisioning renewable energy, excluding imported hydropower, to constitute 35% of the total capacity by 2037. In addition to this, there is an accompanying objective of achieving a 5–6% improvement in energy efficiency, as outlined in the IEA report of 2021a.

**Table 4.** DCE varied value in three alternative.

| Share of RE in 2030 | Type of RE | Electricity Tariff Monthly Increasing |
| --- | --- | --- |
| 10%/15%/25%/35% | Solar/biomass/small-scale hydro | 2%/5%/10%/15%/25% |

Thus, the range of RE share and the increasing electricity tariff were based on the efficiency target. The completed revised draft questionnaire was used to experiment with a similar group of target samples for 30 respondents to confirm the varied value of the alternative in the DCE question. The questionnaires were carried out by enumerators from Thammasart University. All enumerators were trained by the researchers' team before implementing the pre-test and data collection.

### 3.3.4. Blocks and Choice Tasks

We generated the requisite combinations of choice sets using the numerical analysis software MATLAB version 1.0.0. To ensure the quality of responses, we assigned each respondent seven to eight-choice tasks. Research has shown that response quality tends to diminish when individuals are required to make between eight and 16 comparisons (Pearmain and Kroes 1990). Each respondent's set of choice tasks formed a block, with careful configuration to ensure an equal occurrence of alternatives within each block. Thus, the total choice tasks of this questionnaire were 93 choice tasks with 12 blocks, but the respondents did not need to cover all of them. A D-optimal design was created for three alternatives: two hypothetical and one present circumstance. The average number of replies per responder was one block of about 7–8 choice tasks. Respondents were assigned to the blocks in such a way that the same number of people responded to each one (Numata et al. 2021).

### 3.3.5. Regression Analysis

In the previous session, it targeted a survey covering 250 households in Bangkok. To ensure the accuracy of the regression analysis, it excluded households with unusually high or low monthly electricity bills from the sample. The conditional logit model was used to estimate the willingness to pay (WTP) of households in Bangkok, Thailand. In this model, it was assumed that the utility of choosing a particular alternative was a linear function of two factors: the proportion of renewable energy (RE) and its price. We represented different RE sources, such as solar, biomass, hydropower, and wind, and were represented

by dummy variables. The baseline type for comparison in the model was solar, which was considered the current status quo for Thailand. In mathematical terms, for a given respondent denoted as "*i*", the utility associated with selecting an alternative "*j*" can be expressed as a function of the attributes or characteristics of that particular alternative "*j*". The utility function ($U_{ij}$) consists of two distinct components: (1) A deterministic part ($V_{ij}$) that accounts for the observed characteristics of the alternative. (2) A stochastic error component ($\varepsilon_{ij}$) that accommodates unobserved or random variables, introducing a level of uncertainty into the utility calculation.

$$U_{ij} = V_{ij} + \varepsilon_{ij} \tag{1}$$

where the deterministic part, $V_{ij}$, constitutes the measurable portion of utility and is associated with both the attributes of the alternatives and the characteristics of the respondent. This component is represented as a linear-in-parameter function, which can be expressed as follows:

$$V_{ij} = \sum_k X_{jk}\beta_k \tag{2}$$

where $X_{jk}$ is the $k$ attribute value of the alternative $j$, and $\beta_k$ is the coefficient associated with the $k$th attribute.

### 3.3.6. WTP Estimates Analysis

To estimate Willingness to Pay (WTP) for various renewable energy (RE) share levels and different types of RE, we utilized the outcomes obtained from the conditional logit analysis. This involved converting both statistically significant and insignificant parameters into marginal WTP values. This conversion was achieved by dividing the marginal utility associated with attributes by the marginal utility of price. The utility function for households can be expressed as follows:

$$V_j = \beta_1 share_j + \beta_2 Solar_j + \beta_3 Wind_j + \beta_4 Hyd_j + \beta_5 Bio_j + \beta_6 Price_j \tag{3}$$

where $V_j$ is the utility of choice set $j$; $share_j$ is the RE share amongst total electricity production of choice set $j$; $Solar_j$, $Wind_j$, $Hyd_j$, and $Bio_j$ are dummy variables representing RE types of choice set $j$; and $Price_j$ represents the percentage of increasing monthly electricity tariffs.

To examine $Price_j$ at different $share$ levels, we specified $share_j$ and determined the changes in $WTP_j$ using the following function:

$$WTP_j = \frac{\beta_1(share_j - share_{sq}) + \beta_2 Solar_j + \beta_3 Wind_j + \beta_4 Hyd_j + \beta_5 Bio_j}{-\beta_7} \tag{4}$$

In this analysis, we employed the Apollo package in R, which was specifically designed for the estimation and application of choice models. The Apollo package offers a wide range of modeling capabilities, ranging from the basic Logit model to more complex structures that incorporate random coefficients, as detailed in the work by Numata et al. in 2021.

## 4. Results

### 4.1. Demographic Exploration

Based on the sampling technique, the research team collected 250 respondents by applying multi-stage and stratified sampling techniques. Sampled households will be allocated proportionally to zones, districts, and sub-districts, respectively. Finally, the research team will select households randomly starting from the first alley of the road, then select the next households at least 3–5 houses apart.

Table 5 shows the socio-economic profile of the respondents, which consists of gender, occupation, age, education level, income, family members, and the number of members during the day. The respondents in this survey were more female than male, accounting for 65.6 percent and 34.2 percent, respectively. The majority occupation was manager in the workplace at

56.5 percent, followed by self-employed at 14 percent, and the Student/Retired/Unemployed at 10 percent. Most education levels in this survey were primary (28 percent), followed by bachelor's degrees (24.4 percent). The age range of respondents was 51–60 years old at 26.4 percent, while the highest range of income was 15,001,000 THB to 25,000 THB (415$ to 690$) per month. Finally, the average family member was 4 people, while the member who stayed during the day was around 2 people per household.

**Table 5.** Socio-economic Investigation. (n = 250).

| Variables | f | % | Average |
|---|---|---|---|
| Gender | | | |
| Male | 86 | 34.4 | |
| Female | 164 | 65.6 | |
| Total | 250 | 100 | |
| Occupation | | | |
| Unskilled Labor | 14 | 6 | |
| Officer | 16 | 6 | |
| Manager | 141 | 56.4 | |
| Self-employed | 35 | 14 | |
| Skilled-labor | 1 | 0.4 | |
| Housekeeper | 10 | 4 | |
| Student/Retired/Unemployed | 25 | 10 | |
| Others | 8 | 3.2 | |
| Total | 250 | 100 | |
| Education | | | |
| Primary | 70 | 28 | |
| Secondary | 36 | 14.4 | |
| High school | 39 | 15.6 | |
| Vocational College | 38 | 15.2 | |
| Bachelor's degree | 61 | 24.4 | |
| Post Graduate Degree | 6 | 2.4 | |
| Total | 250 | 100 | |
| Age | | | |
| Less than 20 years old | 23 | 9.2 | |
| 21–30 | 32 | 12.8 | |
| 31–40 | 54 | 21.6 | |
| 41–50 | 59 | 23.6 | |
| 51–60 | 66 | 26.4 | |
| More than 60 years old | 16 | 6.4 | |
| Total | 250 | 100 | |
| Income | | | |
| Less than 15,000 THB | 54 | 21.6 | |
| 15,001–25,000 THB | 63 | 25.2 | |
| 25,001–35,000 THB | 42 | 16.8 | |

**Table 5.** *Cont.*

| Variables | f | % | Average |
|---|---|---|---|
| 35,001–45,000 THB | 25 | 10 | |
| 45,001–55,000 THB | 17 | 6.8 | |
| More than 55,001 THB | 49 | 19.6 | |
| Total | 250 | 100 | |
| Family Members | | | |
| number of members | | | 4.188 |
| number of members living during the day | | | 2.43 |
| Number of age over 60 | | | 0.196 |
| Number of children | | | 0.812 |

### 4.2. Electricity Consumption Behavior Analysis

To recognize the electricity consumption behavior, this section provided the details of each activity to imply the electricity consumption behavior of metropolitan residents.

Table 6 shows the electricity consumption behavior of residents in metropolitan areas. All respondents used electricity from the "Metropolitan Electricity Authority (MEA)" in Thailand. For the tariff, the residential electricity tariff was 92 percent, while the remaining was the electricity tariff for commercial activities at 8 percent. In addition, respondents had a small business in their house at 53.2 percent; the major business was the grocery shop (36.09%), followed by the restaurants (16.54%). Based on the total number of business houses, they were all not separate from the meter, which made the proportion of monthly electricity used up to 49.62 percent of half of the total electricity used. Interestingly, all respondents revealed that the average power outage was 3.6 times per year in the metropolitan area.

**Table 6.** Electricity consumption of the households' exploration (n = 250).

| Indicators | f | % |
|---|---|---|
| House Status | | |
| Name on the bill and responsible for this cost | 47 | 18.8 |
| Different name from the bill but respond to this cost | 139 | 55.6 |
| Neither | 64 | 25.6 |
| Total | 250 | 100 |
| Number of the Houses | | |
| 1 house | 246 | 98.4 |
| 2 houses | 3 | 1.2 |
| More than 2 houses | 1 | 0.4 |
| Total | 250 | 100 |
| Use the Metropolitan Electricity Authority (MEA) | | |
| Yes | 250 | 100 |
| No | 0 | 0 |
| Total | 250 | 100 |

**Table 6.** *Cont.*

| Indicators | f | % |
|---|---|---|
| The Electricity Tariff | | |
|     Residential electricity tariff | 230 | 92 |
|     The electricity tariff for manufacturing activities | 0 | 0 |
|     Electricity tariff for commercial activities | 20 | 8 |
|         Total | 250 | 100 |
| Does Your Household Run a Small Business | | |
|     Yes | 133 | 53.2 |
|     No | 117 | 46.8 |
|         Total | 250 | 100 |
| Type of Business | | |
|     Grocery | 48 | 36.09 |
|     Specialty store | 5 | 3.76 |
|     Restaurant | 22 | 16.54 |
|     Coffee shop | 9 | 6.77 |
|     Laundry services/ironing | 4 | 3.01 |
|     Barber/Beauty shop | 11 | 8.27 |
|     Tailor shop | 6 | 4.51 |
|     Hotel/Inn | 1 | 0.75 |
|     Agriculture | 0 | 0.00 |
|     Bike/car wash | 0 | 0.00 |
|     Bike/car repair shop | 8 | 6.02 |
|     Household manufacturing plant | 0 | 0.00 |
|     Others | 19 | 14.29 |
|         Total | 133 | 100 |
| Your Business Separate Meter | | |
|     Yes | 0 | 0 |
|     No | 133 | 100 |
|         Total | 133 | 100 |
| The Proportion of Monthly Electricity Used From the Business | | |
|     A quarter | 15 | 11.28 |
|     A half | 66 | 49.62 |
|     Three quarter | 20 | 15.04 |
|     Almost all | 32 | 24.06 |
|         Total | 133 | 100.00 |
| The Average of Power Outage Per Year | 3.676 | |

Figure 3 displays the average electricity consumption; this data were obtained from the electricity bill, which all respondents consented to show the team of researchers following the human ethics protocol. Of all households surveyed in the metropolitan area, most respondents consume 100–200 KWh of electricity per month.

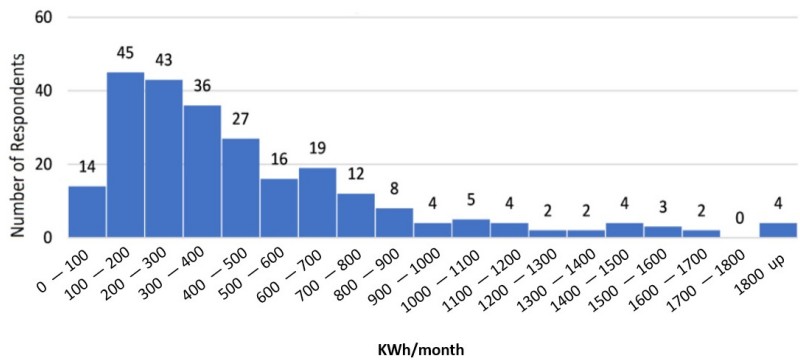

**Figure 3.** Monthly Electricity Consumption.

### 4.3. Attitude and Opinion toward Renewable Energy

This section was related to the interest in the specific issue; therefore, it was divided into two sub-sections: the attitude toward renewable energy and the opinion on the management of renewable energy in Thailand.

Figure 4 shows that the respondents knew the different types of renewable energy (Solar PV at 96%, Wind Power at 78%, Biomass at 58.4%, and Hydro Power at 72.4%). When considering the eco-friendly feeling of each sort of renewable energy, solar cells had a very eco-friendly feeling at 48.4 percent and wind power at 43.6 percent. For biomass, it was at a not-certain level of eco-friendliness at 36.2 percent. The hydropower was still in the eco-friendly category at 36.8 percent.

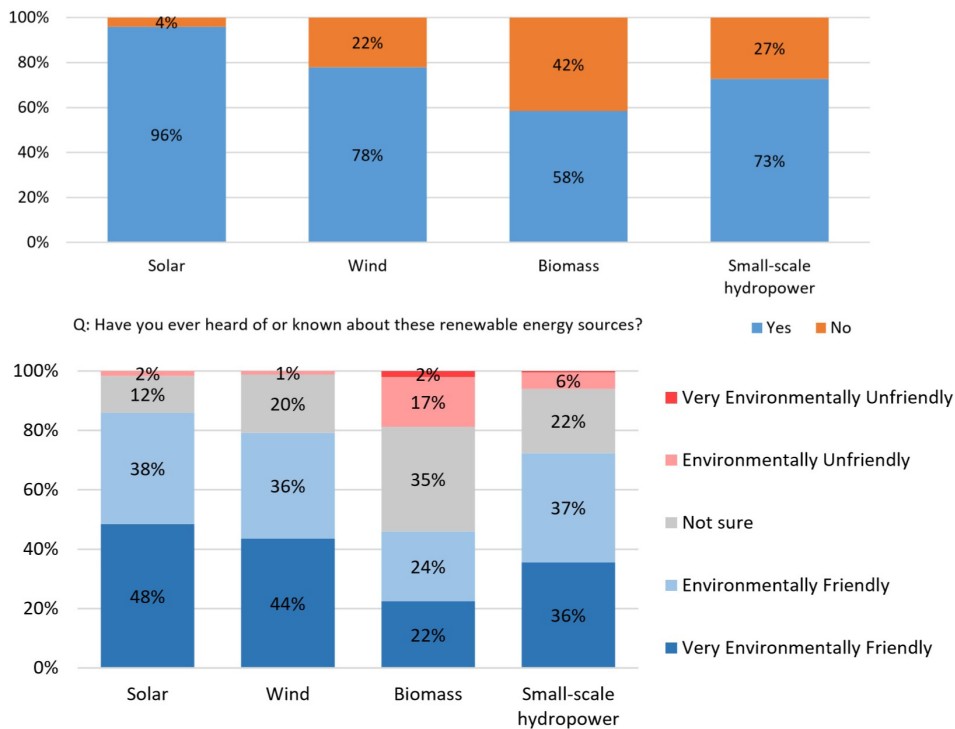

**Figure 4.** Attitude toward RE.

Interestingly, Table 7 shows that all respondents believed that the household's electricity bill would increase if the share of renewable energy rose to 75.6%. Renewable energy machines are usually calculated based on their maintenance and repair costs.

**Table 7.** Households' Attitude toward Renewable Energy. (n = 250).

| Items | f | % |
|---|---|---|
| Do you think that your household's monthly electricity bill would increase if the share of renewable energy increases? | | |
| Yes | 189 | 75.6 |
| No | 61 | 24.4 |
| Total | 250 | 100 |

From the past to the present, environmental politics in Thailand have been highly controversial. Figure 5 displays the respondents' opinions about the administration and management of renewable energy in metropolitan areas. The opinion that the government should provide low electricity costs to poor households was strongly agreed upon at 68.4 percent, while the opinion that the government should provide electricity at a higher price to encourage electricity-saving practices was also strongly disagreed upon at 54 percent. Interestingly, the opinion that they do not care about the source of power; they care only about the price was found to be a similar ratio between strongly (30.4%) agree and disagree (28%). Furthermore, 33.2 percent of respondents agreed that they would be willing to pay more if blackouts decreased. The respondents understand that providing a good quality of power is under the responsibility of the Government, it is the welfare and right to have a quality power service. Why do they need to pay more for having a good one.

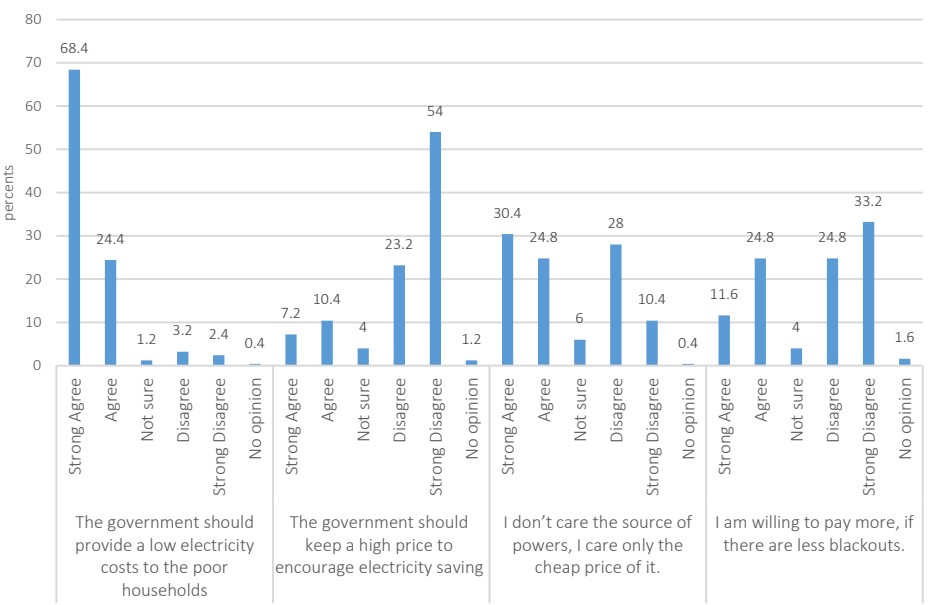

**Figure 5.** Households' opinion on the administration and management of renewable energy in Thailand.

*4.4. Cognition on Environmental Issues and Climate Change*

This section was focused on current environmental issues and climate change. It was measuring the perception, understanding, and cognitive level of a global issue. As a result, this measure may be relevant to the decision-making process on WTP for renewable energy.

Figure 6 illustrates the primary environmental concern within metropolitan areas. Respondents were asked to vote twice on a list of environmental issues. The top-ranked concern was air pollution, followed by solid waste management as the second most significant environmental issue.

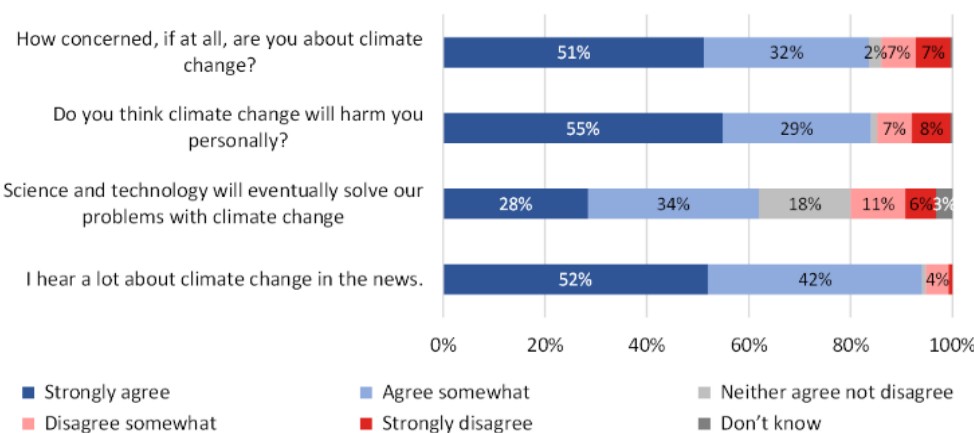

**Figure 6.** Opinion on Environmental Issues.

Figure 7 provides insights into the cognitive perception of climate change among respondents. Notably, 51 percent strongly agreed that they hold a high level of concern regarding climate change, while 55 percent believed it would negatively impact their quality of life. This heightened awareness can be attributed to the fact that 52 percent of respondents strongly agreed that they receive substantial news and information about climate change. Additionally, it is noteworthy that 33.6 percent of all respondents held the belief that science and technology will ultimately resolve the climate change issue, albeit at an agreed level. This perspective is likely to drive further interest in Carbon Dioxide Removal (CDR) technology, an emerging field for pollution control in urban areas. It is worth mentioning that the survey instrument included questions pertaining to CDR, and these findings will be subject to future analysis.

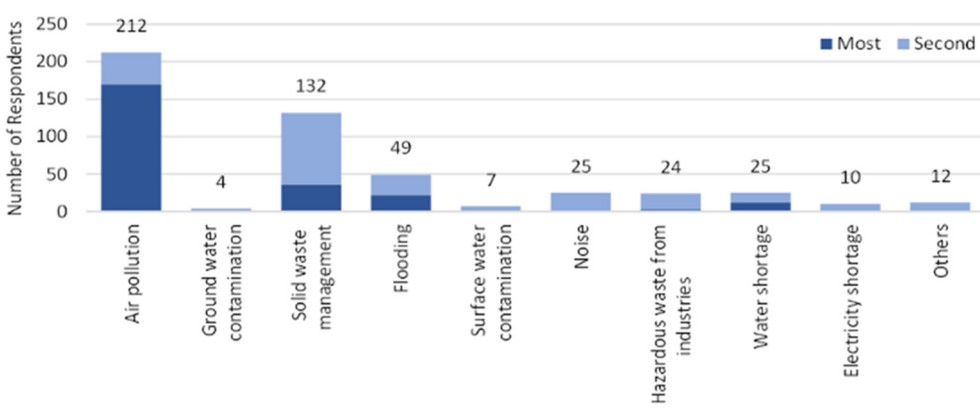

**Figure 7.** Cognition on Climate Change.

*4.5. Discrete Choice Experiment Analysis*

To identify people's "stated preference" for renewable energy, the discrete choice experiment (DCE) was used to analyze their willingness to pay. The contingent valuation method involves directly inquiring about the value of the environment. On the other hand, the Discrete Choice Experiment (DCE) method, also relying on questionnaires, assesses the value of the environment by soliciting preferences for various alternatives related to the utilization of power generated from different sources of renewable energy. Respondents were asked which of several alternatives they preferred.

Table 8 shows that respondents preferred alternative C from the total of all blocks holistically. The situation described above represents the current status quo in Thailand concerning the use of renewable energy. It means that overall, people in the metropolitan area are still unwilling to pay for and change for improving renewable energy. However, when considering each choice task (CT), there were some CTs that respondents voted to

accept as a different alternative from the status quo. Those selected CTs will be expressed in Table 9 to show the preference values of respondents in Bangkok.

**Table 8.** Discrete Choice Experiment of Households for Renewable Energy.

| Discrete Choice | Alternative A (f) | Alternative B (f) | Alternative C (Status Quo) (f) |
|---|---|---|---|
| Block 1: CT 1–8 | 38 | 37 | 101 |
| Block 2: CT 9–16 | 38 | 37 | 101 |
| Block 3: CT 17–23 | 37 | 46 | 71 |
| Block 4: CT 24–32 | 20 | 36 | 91 |
| Block 5: CT 33–40 | 26 | 39 | 82 |
| Block 6: CT 41–48 | 42 | 36 | 62 |
| Block 7: CT 49–56 | 34 | 21 | 92 |
| Block 8: CT 57–64 | 23 | 32 | 92 |
| Block 9: CT 65–72 | 16 | 37 | 87 |
| Block 10: CT 73–80 | 13 | 30 | 97 |
| Block 11: CT 81–88 | 44 | 44 | 52 |
| Block 12: CT 89–92 | 25 | 17 | 98 |

**Table 9.** The results of the utility function estimates.

| Variable | Study Area (Bangkok) |
|---|---|
| Price | −0.118 *** |
| (% of the monthly bill) | (0.007) |
| RE Share (%) | 0.015 *** |
| | (0.005) |
| Renewable Energy Types | Solar |
| Solar | - |
| Biomass | −0.361 *** |
| | (0.114) |
| Hydropower | −0.337 *** |
| | (0.114) |
| Wind | −0.272 ** |
| | (0.112) |
| ASC (SQ) | 0.174 |
| | (0.126) |
| Obs | 5691 |
| Number of Households | 250 |
| Long-likelihood | −1645 |

ASC is an alternative-specific constant, RE is the renewable energy. Note: Robust standard errors are in parentheses. *** and ** indicate statistical significance at the 1% and 5% levels, respectively.

The table below presents the outcomes of econometric models using a conditional logit model (Table 9). In this model, only the attributes of the alternatives were considered independent variables. It also incorporated alternative-specific constants (ASCs) to assess the influence of the current status quo. A positive ASC coefficient suggests that respondents tend to favor an increase in the share of renewable energy compared to its current level.

Table 9 shows that the overall data of the households in Bangkok was significant statistically. The households in Bangkok prefer the increased price (0.007) negatively at the statistical significance level of 1%. It means that the higher the price, the lower the utility of households. For the coefficient of higher RE shares, it was a positive (0.005) at the statistical significance of 1%. The utility result was used to calculate WTP further.

As depicted in Table 10, when examining the estimation of the mean Willingness to Pay (WTP) as a percentage of monthly electricity bills in US dollars while increasing the share of renewable energy (RE) to various levels, households tend to exhibit a preference for a higher proportion of renewables within the electricity mix. Notably, among the various types of renewable energy sources, solar energy consistently stands out with the highest WTP values. For instance, when the RE share reached 40%, the WTP values for solar cells reached 5.54%. Biomass energy was valued the lowest.

**Table 10.** Percentage estimates of WTP for various types of resources portion of monthly electricity bill.

| Thailand | RE Share | Solar % of Monthly Electricity Bill (USD) | Biomass % of Monthly Electricity Bill (USD) | Hydropower % of Monthly Electricity Bill (USD) | Wind % of Monthly Electricity Bill (USD) |
|---|---|---|---|---|---|
| | 20% | 2.92% (2.33) | −0.14% (−0.12) | 0.06% (0.05) | 0.61% (0.49) |
| (status quo = 9%) | 30% | 4.23% (3.38) | 1.17% (0.93) | 1.37% (1.10) | 1.92% (1.54) |
| | 40% | 5.54% (4.43) | 2.48% (1.98) | 2.68% (2.15) | 3.24% (2.59) |

## 5. Discussions

The empirical results of socio-economic exploration found a diversity of residential status in Bangkok. The Metropolitan Electricity Authority (MEA) was the main and only organization to provide electricity to households in Bangkok. It seems the state has a monopoly, in which people do not have choices for electricity service and still face power outages 3.6 times per year on average. It was not surprising why the household respondents have a positive attitude toward RE as there are more choices (Solar Cell 96%, Wind Power 78%, Biomass 58.4%, and Hydro Power 72.4%), even if they realize that their electricity bill will increase. Also, the household respondents have the opinion that providing a good quality of power is under the responsibility of the Government, it is for the welfare and the right to have a quality power service. Finally, the households in metropolitan areas have cognition on environmental problems such as air pollution and climate change (agree level at 33.6%); however, they still chose to use the existing status of electricity production and remain unchanged to apply RE generally.

Earlier research conducted among Bangkok residents showed little support for the ban on burning solid fuels and oil. The Thai government endeavored to address this pattern by introducing the Alternative Energy Development Plan (AEDP), which aimed to boost the renewable energy (RE) portion to 36% and incorporate nuclear power into the 20-Year Power Development Plan 2010–2030, as outlined in Malahayati's 2020 report. To increase the adaptation of RE at the household level, the Government needs to support the specific source of households' preferences. Even if the share of electric power was not much, solar power was still the highest preference among others; the WTP values of solar cells were 5.54%. It can increase gradually to the bottom-line RE use, pushing toward the national level of RE use, respectively.

Moreover, as there have been a few studies on WTP for RE in Thailand (Suanmali et al. 2018; Ministry of Energy 2015), this present research has increased the study for energy-saving and environment-friendly that the household in the metropolitan area, which was believed to be the potential group of people because of the RE information and accessibility, is still unchanged to apply RE. However, the result shows that the households in Bangkok prefer the increased price (0.007 significant level). It means that increased prices reduced the utility for households. In addition, the higher RE share was a positive result (0.005 level) for the willingness to pay for the RE adaptation. The results of this study can be beneficial for the government to plan further initiatives to encourage RE among residents.

This article used a method and composition similar to the WTP for Renewable Energy in Myanmar Case (Numata et al. 2021) under the project collaboration between the University of Tokyo and ASEAN universities. Thus, the different results will be highlighted to express new findings. Based on the WTP estimation, Myanmar prefers to increase the share of RE from a diversity of sources but does not prefer to increase the price of electricity. While the households in Bangkok prefer the increased price significantly and the willingness to pay for the RE adaptation. The reason to support this finding is based on the attitude and opinion investigation of the respondents in Bangkok. They believed that the household's electricity bill would increase if the share of renewable energy rose. Also, respondents in Bangkok do not care about the diversity of RE shares; they care only about the increasing price. Thus, the main hypothesis was accepted: people in Bangkok are willing to invest in renewable energy.

## 6. Conclusions

The finding shows a diversity of socio-economic status among residents in Bangkok in the use of electricity for their living and small businesses with different tariffs. Households in Bangkok have a positive attitude toward and are willing to pay for RE, including solar cells, wind, and hydropower, except for biomass, as they are not sure of its level of eco-friendliness. They all have a unique opinion that providing a good quality of power is under the responsibility of the Government, as it is for the welfare and the right to have a quality power service, so they do not need to pay more for a better one. Even so, the majority of households understand that their electricity bill will increase if the share of RE increases. For the cognition of environmental problems, households in metropolitan areas voted for air pollution first, while climate change also had a high concern because of the frequency of news and press publications. They expect that CDR technology can solve climate change, even if they have just realized and understood it. Finally, the overall household in Bangkok still uses the existing status of electricity production based on the provided alternatives. The majority of Bangkok households were strongly opposed to the price increase. However, there were some alternatives that respondents voted to accept, including REsources, costs, and shares, apart from the status quo. The preference for applying RE alternatives is solar energy (5.54% even though the price will increase), while biomass is the least willing to pay at 2.48% when the RE share is up to 40% among the different alternatives of RE share in the electricity mix. This research endeavor represents a collaborative effort between the University of Tokyo and several universities within the ASEAN region. This study builds upon prior investigations that had previously disclosed findings related to Willingness to Pay (WTP) in the context of Myanmar. However, it is noteworthy that the examination of the Thai case yielded distinctive insights, notably indicating an augmentation in academic contributions. Furthermore, the findings and analytical outcomes derived from this study serve as essential foundational data that can be strategically applied by governmental authorities in shaping Renewable Energy (RE) policy initiatives.

Some limitations affected this research operation, including the collection of data during the COVID-19 pandemic and the inability of some respondents to find previous electricity bills to analyze household consumption behavior. Another limitation of the WTP method is the external factor. When external factors can influence the value, this measure becomes less precise. For instance, in a distressed sale, the price may be lowered due to urgency, but this does not necessarily reflect the actual value of the item. Sometimes, a renewable energy source may have unique features that are especially attractive to a buyer, leading them to pay more than the market average, which is called a premium. Thus, the equation used to measure the WTP has to be careful when including and excluding factors for analysis. Future studies should be applied to other potential groups of respondents, such as urban and suburban households in different parts of Thailand; this would increase the scope of the grounded study and can be generalized for renewable energy. In addition, WTP is an economic analysis that realizes the win-win situation among the public and private

sectors as the power generators and households as the users to increase RE sustainability. A future synergistic study of multi-sectors for applying RE is therefore recommended.

**Author Contributions:** Conceptualization, V.N., M.S., M.N. and D.d.B.A.; methodology, S.J., V.N., O.T. and T.W.; investigation, S.J., V.N., O.T. and T.W.; writing—original draft preparation, S.J.; writing—review and editing, V.N., S.J., M.S., M.N. and D.d.B.A.; supervision, V.N., M.S., M.N. and D.d.B.A.; project administration, S.J., V.N., O.T. and T.W. All authors have read and agreed to the published version of the manuscript.

**Funding:** This research was funded by the Economic Research Institute for ASEAN and East Asia (ERIA).

**Institutional Review Board Statement:** Ethical review and approval were waived for this study due to this study was not a medical study, no risk of disadvantage to the vulnerable population, and provided the consent form for all respondents.

**Informed Consent Statement:** Signed informed consent was obtained from all respondent in this survey.

**Data Availability Statement:** The data presented in this study are available on request from the corresponding author. The data are not publicly available due to ethics restrictions.

**Acknowledgments:** We would like to express our gratitude to the team of the University of Tokyo for the great research collaboration. Also, we would like to thank the Asian Institute of Technology and Thammasart University for supporting the research project implementation.

**Conflicts of Interest:** The authors declare no conflict of interest.

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
