# Peer review of "Households’ Willingness to Pay for Renewable Energy Alternatives in Thailand"

_socsci, doi:10.3390/socsci12110634_

Round 1

Reviewer 1 Report

Comments and Suggestions for Authors

1.      This paper provides an empirical study of the WTP for renewable energy alternatives in Thailand.  If people want to know the situation in Thailand, then this paper can be a useful reference.  However, some fundamental issues remain to clarify or improve.  It is also not clear why in Table 9 on p.17 there is a statement of the Philippines even though this paper uses the Thai data.  The research methodology and steps remain not clear, too.

2.      The mathematical expressions in this paper are difficult to read and need corrections.  For instance, on page 10 the ij term should be in subscripts.  Currently the ij terms are messed up with the variables.  The first equation on page 10 should be a linear regression.  It should be clearly expressed and explained as a linear regression.

3.      Because there are subscripts ij in all variables on page 10.  This paper should use a panel data regression.  However, it is not clear whether or not a panel data regression is indeed used for estimation because there is no explicit report of the panel data estimation and the panel data statistical tests.

4.      It is not clear why in Table 9 on p. 17 it mentions “The ASC for the Philippines would be perfectly correlated…” This paper deals with the Thai data.  It is not clearly why the Philippines suddenly showed up in the text.  Was this sentence copied and pasted from somewhere else with the Philippines data? If so, this way should not be encouraged in doing research.

5.      The terms ‘Where’ under an equation on pages 7 and 10 should be corrected as ‘where’ and aligned to the very left.

6.      This paper actually uses conditional Logit regression for estimation.  The descriptions and justifications for using conditional Logit regressions should be further elaborated.  The related statistical tests should be done and reported.

7.      Table 9 reports the estimation results of a utility function.  However, it is not clear how the utility function is constructed and why renewable energy types look like consumptions in the utility function.  In addition to the econometric problems, the authors should explain and justify the construction of the utility function before the estimation of the utility function.

8.      There is no research hypothesis constructed in this paper.  Research hypotheses should be constructed from theories or literature.

9.      The reference formats are not in the MDPI journal formats.  They should be adjusted.

Comments on the Quality of English Language

The revised manuscript should be professionally edited by a native English speaker before re-submission.

Author Response

Comments

Response

1. This paper provides an empirical study of the WTP for renewable energy alternatives in Thailand.  If people want to know the situation in Thailand, then this paper can be a useful reference.  However, some fundamental issues remain to clarify or improve.  It is also not clear why in Table 9 on p.17 there is a statement of the Philippines even though this paper uses the Thai data.  The research methodology and steps remain not clear, too.

Revised, due to this research is the collaboration between Japan and Universities in ASEAN. I have cut it out the referring to the Philippine data.

2. The mathematical expressions in this paper are difficult to read and need corrections.  For instance, on page 10 the ij term should be in subscripts.  Currently the ij terms are messed up with the variables.  The first equation on page 10 should be a linear regression.  It should be clearly expressed and explained as a linear regression.

-Corrected the equation based on your comments and Revised the data analysis by starting with the linear regression

 3. Because there are subscripts ij in all variables on page 10.  This paper should use a panel data regression.  However, it is not clear whether or not a panel data regression is indeed used for estimation because there is no explicit report of the panel data estimation and the panel data statistical tests.

-This study is the collaboration between the University of Tokyo and Universities in ASEAN. We made an agreement to use the linear regression as the common analysis for all cases. Moreover, we found from the literature that WTP mostly applied linear regression. Thus, we rebuttal to this point of applying the panel data regression

4. It is not clear why in Table 9 on p. 17 it mentions “The ASC for the Philippines would be perfectly correlated…” This paper deals with the Thai data.  It is not clearly why the Philippines suddenly showed up in the text.  Was this sentence copied and pasted from somewhere else with the Philippines data? If so, this way should not be encouraged in doing research.

Revised, due to this research is the collaboration between Japan and Universities in ASEAN. I have cut it out the referring to the Philippine data.

5. The terms ‘Where’ under an equation on pages 7 and 10 should be corrected as ‘where’ and aligned to the very left

Revised the format based on the comment

6. This paper actually uses conditional Logit regression for estimation.  The descriptions and justifications for using conditional Logit regressions should be further elaborated.  The related statistical tests should be done and reported

-Revised, the result of conditional Logit regressions will be used to analyze the WTP estimation. I have revised the explanation in the data analysis session

7. Table 9 reports the estimation results of a utility function.  However, it is not clear how the utility function is constructed and why renewable energy types look like consumptions in the utility function.  In addition to the econometric problems, the authors should explain and justify the construction of the utility function before the estimation of the utility function

-Revised by adding the introduction of the table 9.

8. There is no research hypothesis constructed in this paper.  Research hypotheses should be constructed from theories or literature

-Revised by adding more literatures for creating the social science hypothesis in 2.3 session

9. The reference formats are not in the MDPI journal formats.  They should be adjusted.

Revised,

Reviewer 2 Report

Comments and Suggestions for Authors

Generally, the descriptions of methods should be improved. More comments in the file.

Author Response

Reviewer 2

Comments

Revision

1. The paper and methodological approach could be interesting for readers and policymakers. The aim of research is described. What is the aim of the paper? Are these aims the same

Yes it is the same, Revised already

2. Generally, the description of the methods and analytical part should be improved. The tools, methods and conclusion should be described more clearly and directly.

Revised, the structure of method session has been changed to be more clear understanding

3. The used method and composition of the article is very similar to Numata et al. Thus, it should be strongly highlighted what is new in this paper because Numata’s study describes analogous problem for a country other than Thailand, thus in the Discussion section there should be the results coming from two analogous research compared.

Revised, I added the last paragraph in the discussion session to show the different finding

4. The advantages and disadvantages of used methods should be described. I think each researcher after conducting the study has an opinion about limits of the used methods.

Revised, I added the disadvantage of applying the WTP in the conclusion session

5. The literature review should be deeper. The background of the used method is the consumer choice theory. There are some methods to estimate the utility function and WTP. For example, the description of each good by attributes (features, characteristics) is  presented in publications such as:

·         Hanley N., Wright R.E., Adamowicz V. Using Choice Experiments to Value the Environment, Environmental & Resource Economics 11, 1998.

·         Louviere J. J., Flynn T. N., Carson, R. T. Discrete choice experiments are not conjoint analysis. Journal of Choice Modelling, 2010.

·         Lancaster, Kelvin J.A New Approach to Consumer Theory. Journal of Political Economy 74, no.2, 1966.

·         Scarpa R., Willis K., Willingness-to-pay for renewable energy: Primary and discretionary choice of British households' for micro-generation technologies, Energy Economics 32.

·         Minnis A. M., Agot K., Weinrib R., Young Women’s Stated Preferences for Biomedical HIV Prevention: Results of a Discrete Choice Experiment in Kenya and South Africa, Acquir Immune Defic Syndr No 4, 2019.

Revised, I added the mentioned literature to the section 2.2 and 2.4

6. Section 3 about the used method, tools and the research process should be improved. Especially, the part describing task choices and blocks because under Table 1 there is some mix between the literature review and the methods used in this paper

-Revised the structure of section 3 to be more clear for the reader 

7. The description of methods used by other authors should be presented, but e.g. I would like to find out about the methods and tools as well as results of simulation conducted by the Authors of reviewed article. For example, the description of the number of choice tasks in the Authors’ research. Numata et al described the used tools and function more deeply. Thus, their description is clear for the reader. Here, the description is not clear. If  the reader does not know the Numata's and their colleagues' paper, he/she could not understand this description

Revised, I added the mentioned literature to the section 2.4

8. It will be clearer if e.g. one combination of the alternatives of the choice tasks was characterized. Are the choice tasks, the attributes and the set tasks the same like Numata et al?

-Revised by combining it in the chapter 3

9. Table 1 presents the alternative choice tasks with their attributes. What is the difference between scenario and alternative (please look at Numata) or the scenario in the reviewed paper and the levels of attributes (please look at Minis et al, 2019)?

Revised to be the same as Numata by applying the “alternative”

10. The use of different words (scenarios, attributes, attribute levels (line 215) for the same aspects is misleading. Maybe, in the footnote, it should be explained that these words are the synonyms in this context. The source of Table 1 is Numata. This is written in the reviewed paper but in Table 1 there was one change introduced - the percentage of RES.   

11. The description of the model should be more precise. 

Revised by adding more explanation before the model of table 10

12. Step 4 (line 143) should be described more deeply. It is the heart of analyzed step of the research. 

Revised, by modifying the paragraph below to show how to analyze WTP

13. Under formula in line 213 values for each parameter (n, t, a, c) should be presented in this research. In line 273 and 282 the i, j are the subscripts and they should be written in a different way. The formula in line 282 is not complete because the Σ should have subscripts.

Revised

14. What is ‘f’ in the table 5. Is this the number of women?

Frequency of respondent

15. How was the analysis of monthly electricity costs according to the respondents’ electricity billing made? (1) Was the analysis made on the base of the annual electricity costs?  (2) Have the respondents (households) given sample bill from one month? If the analysis was made according to (2), the results (costs) are incomparable and the analysis (Figure 3) does not fall under the methodological rigour – the collected data are incomparable

Revised, we asked respondents to give the current bill for analyzing the average consumption. To prevent the missed understanding, I cut the comparison out.  

16. In line 376 the sentence is not logical because of the word ‘also’.

Revised.

17. The description in lines 395 -398 should be more detailed. It is not clear if there were 2 rounds of survey and if the question has only one choice or more.

Revised.

18. The descriptions of Figure 6 and 7 are confusing.

Revised the explanation of figure 6 and 7

19. What is the relationship between Table 1 and Table 8. Why does Table 8 present scenario A. B, C? What is (f)?

Table 1 was changed to be table 4, it is the example of choice task for respondent. While Table 8 is the result of frequency who vote for.

20. Why is “The ASC for the Philippines” under Table 9? Are the results for Bangkok or Philippines?

Revised, due to this research is the collaboration between Japan and Universities in ASEAN. I have cut it out the referring to the Philippine data.

21. The title of the paper is related with WTP, but in the Conclusion section there is not much about the WTP.

Revised the conclusion by adding the WTP result

22. What IT tools were used for the statistical analysis?

Revised at the end of section 3.

23. The ‘described choice experiment’ is written in a different way in different parts of the paper. 

Revised by clarifying in the section 2.2

24. In line 96: It should be FIT not FIP.

Revised

25. The points 2.1 and 2.2 have the same names.

Revised

26. What is CBD in Table 2?

Revised by adding the note under the table

27. In line 167: Whose are the 90 choice tasks? Yours or Danne's ?

Revised

28. In line 172-173: Whose are the 93 choice tasks? Yours or Numata's? I checked the paper by Numata et al and there were 86 choice sets.

Revising it by adding that “Thus, the total choice tasks of this questionnaire were 93 choice tasks with 12 blocks,”

29. The formulas should be numbered.

Revised.

30. In line 43 Electricity is not a renewable source.  It can be made from RES.  Electricity is not primary energy but secondary (derivative).

Revised.

Reviewer 3 Report

Comments and Suggestions for Authors

Dear authors, thanks for submitting the manuscript. The topic of the paper is quite interesting, and the paper overall is well-prepared and well-structured. at the same time, some recommendations can be given to improve the quality of the paper:

Introduction: state the research questions/hypotheses of this paper;

2. Literature: Please revise the division of section 2 into sections 2.1 and 2.2 (page 2) as the titles match:

2.1. Willingness to Pay Concept and Techniques

2.2. Willingness to Pay Concept and Techniques

Conclusion: Mention and explain the main contributions to the existing literature; identify the ways of implementing the results of this study by businesses and policy-makers, etc.,

Table and figures: Please check if the relevant references/sources are included.

Author Response

Comments

Revision

1. Introduction: state the research questions/hypotheses of this paper;

Revised by the RQ and hypothesis at the last paragraph of introduction section

2. Literature: Please revise the division of section 2 into sections 2.1 and 2.2 (page 2) as the

·         2.1. Willingness to Pay Concept and Technique

·          2.2. Willingness to Pay Concept and Technique itles match

Revised by restructuring the heading of literature.

3.Conclusion: Mention and explain the main contributions to the existing literature; identify the ways of implementing the results of this study by businesses and policy-makers, etc.,

Revised, by adding more explanation of academic and practical contribution to the end of the first paragraph in this section.

4. Table and figures: Please check if the relevant references/sources are included

Revised

Round 2

Reviewer 1 Report

Comments and Suggestions for Authors

The authors have adopted all suggestions from this reviewer and made detailed revisions in accordance. 

Comments on the Quality of English Language

The English writing in the current form is fine.  Of course, a careful proofreading of the final version is always a must.